# Whole body PD-1 and PD-L1 positron emission tomography in patients with non-small-cell lung cancer

A.N. Niemeijer[1], D. Leung[2], M.C. Huisman[3], I. Bahce[1], O.S. Hoekstra[3], G.A.M.S. van Dongen[3], R. Boellaard[3], S. Du[2], W. Hayes[2], R. Smith[2], A.D. Windhorst [3], N.H. Hendrikse[3], A. Poot[3], D.J. Vugts[3], E. Thunnissen[4], P. Morin[2], D. Lipovsek[2], D.J. Donnelly[2], S.J. Bonacorsi[2], L.M. Velasquez[2], T.D. de Gruijl [5], E.F. Smit[6] & A.J. de Langen[1,6]

PD-L1 immunohistochemistry correlates only moderately with patient survival and response to PD-(L)1 treatment. Heterogeneity of tumor PD-L1 expression might limit the predictive value of small biopsies. Here we show that tumor PD-L1 and PD-1 expression can be quantified non-invasively using PET-CT in patients with non-small-cell lung cancer. Whole body PD-(L)1 PET-CT reveals significant tumor tracer uptake heterogeneity both between patients, as well as within patients between different tumor lesions.

[1] Department of Pulmonary Diseases, Cancer Center Amsterdam, Amsterdam UMC, Vrije Universiteit Amsterdam, De Boelelaan 1117, 1081 HV Amsterdam, The Netherlands. [2] Bristol-Myers Squibb Research and Development, Route 206 & Province Line Rd., Princeton, NJ 08543, USA. [3] Department of Radiology and Nuclear Medicine, Cancer Center Amsterdam, Amsterdam UMC, Vrije Universiteit Amsterdam, De Boelelaan 1117, 1081 HV Amsterdam, The Netherlands. [4] Department of Pathology, Cancer Center Amsterdam, Amsterdam UMC, Vrije Universiteit Amsterdam, De Boelelaan 1117, 1081 HV Amsterdam, The Netherlands. [5] Department of Medical Oncology, Cancer Center Amsterdam, Amsterdam UMC, Vrije Universiteit Amsterdam, De Boelelaan 1117, 1081 HV Amsterdam, The Netherlands. [6] Department of Thoracic Oncology, Netherlands Cancer Institute, Plesmanlaan 2, 1066 CX Amsterdam, The Netherlands. Correspondence and requests for materials should be addressed to A.Langen. (email: j.d.langen@nki.nl)

PD-(L)1 immune checkpoint inhibitors have changed the treatment paradigm of patients with advanced non-small cell lung cancer (NSCLC). Durable responses are seen in ~20% of treated advanced NSCLC patients and reported 3-year survival is 17% in the second line setting[1]. PD-L1 expression by immunohistochemistry (IHC) correlates with response and (progression-free) survival following PD-(L)1 monoclonal antibody (mAb) treatment[2–4]. Approximately 10% of the patients respond favorably to PD-(L)1 therapy, despite negative PD-L1 expression as determined by IHC[4,5]. These unexpected responses may be related to heterogeneity of PD-L1 expression within tumors[6,7], such that sampling (false negative) errors limit the predictive value of PD-L1 expression based on IHC. Thereby, it is controversial whether PD-1 expression in tumors can serve as a biomarker for PD-(L)1 mAb treatment[8,9]. Preclinical positron emission tomography (PET) studies with $^{18}$F-BMS-986192[10], an $^{18}$Fluor-labeled anti-PD-L1 Adnectin[11], and $^{89}$Zirconium-labeled nivolumab[12] ($^{89}$Zr-nivolumab) successfully demonstrated non-invasive imaging of PD-(L)1 expression of tumors and may allow to assess inter- and intra-tumoral heterogeneity.

Here we report the first-in-human study results of whole body PET imaging with $^{18}$F-BMS-986192 and $^{89}$Zr-Nivolumab[12] in patients with advanced NSCLC, prior to treatment with nivolumab. Tracer uptake was heterogeneous both between patients, as well as within patients between different tumor lesions. $^{18}$F-BMS-986192 uptake in tumor lesions, measured as SUV$_{peak}$, correlated with tumor PD-L1 expression, measured by IHC. $^{89}$Zr-nivolumab uptake correlated with PD-1 positive tumor-infiltrating immune cells. We also demonstrate a correlation between tumor tracer uptake and response to nivolumab treatment for both tracers. These findings suggest that $^{18}$F-BMS-986192 and $^{89}$Zr-nivolumab PET-CT may be useful imaging biomarkers to non-invasively evaluate PD-1 and PD-L1 expression.

## Results

**Both tracers show favorable distribution for tumor imaging.** Thirteen patients (Supplementary Table 1) with advanced NSCLC consented to undergo three PET scan acquisitions (Fig. 1a). Injection of both tracers was safe with no grade ≥3 tracer-related adverse events (Supplementary Table 2). $^{89}$Zr-Nivolumab distribution from intravascular to extravascular compartments was slow, with increasing tissue accumulation and decreasing concentration in the blood pool over time (Supplementary Figure 1a). Biodistribution was evaluated for various organs (Supplementary Table 3). For both tracers, high tracer accumulation was observed in the spleen, likely due to binding to PD-1 and PD-L1 receptors on lymphocytes and dendritic cells and in the liver, likely due to catabolism of the tracers. In addition, the $^{18}$F-BMS-986192 tracer demonstrated biliary and renal excretion. $^{89}$Zr-Nivolumab demonstrated gastrointestinal excretion typically seen for mAbs. Low to moderate uptake was seen in lung and bone marrow. No tracer accumulation was observed in normal brain for either tracer. Slight uptake in the hypophysis was seen for the $^{18}$F-BMS-986192 tracer.

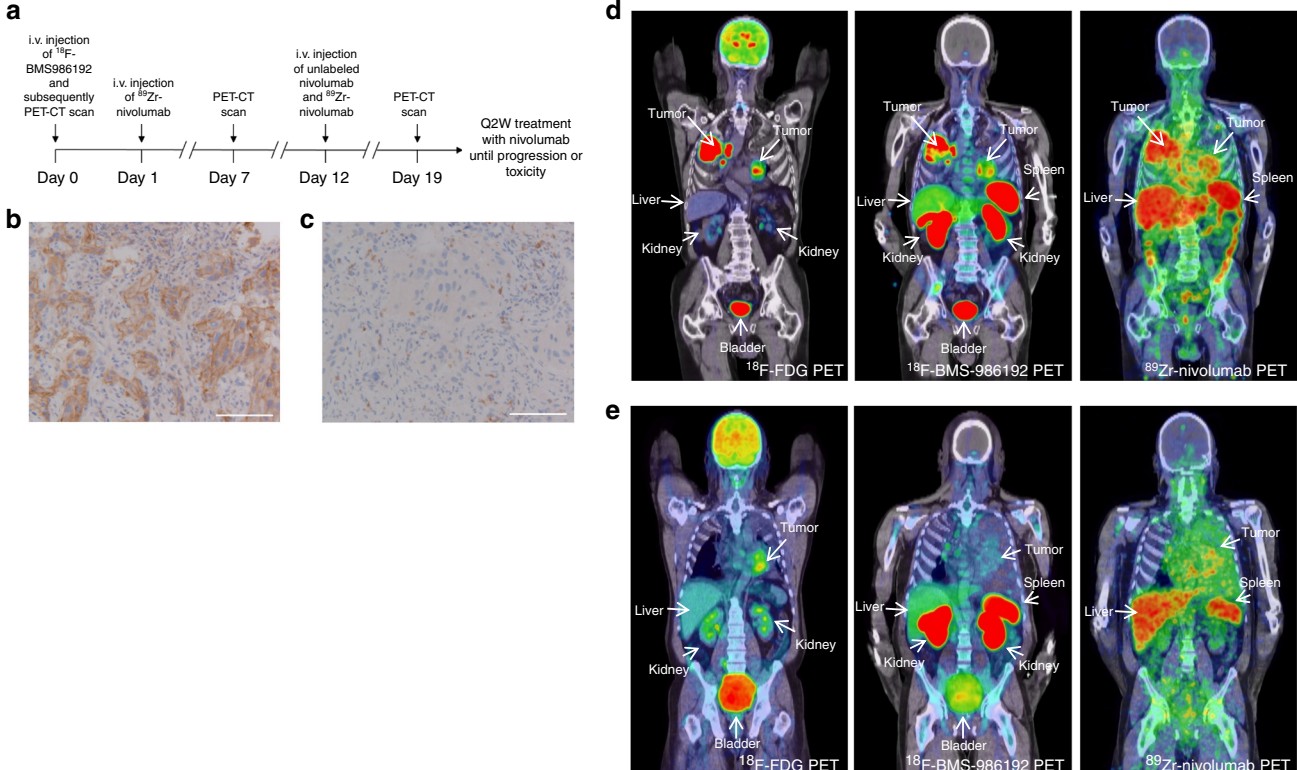

**Fig. 1** Study design, immunohistochemical staining of patient 2, and PET-images of patient 2 and 3. **a** Study design. **b** Immunohistochemical staining of PD-L1 in patient 2. Biopsy of the tumor in the left lower lobe. PD-L1 expression is expressed in 95% of the tumor cells. Scale bar, 100 μm. **c** Immunohistochemical staining of PD-1 in patient 2. PD-1 expression in aggregates was scored as IC1. Scale bar, 100 μm. **d** $^{18}$F-FDG PET (225 MBq) ($^{18}$F-FDG PET scan images of both patients were used from archival PET-scans) demonstrates high glucose metabolism of tumors in both lungs and mediastinal lymph nodes. $^{18}$F-BMS-986192 PET (145.7 MBq, imaging time point 1 h post-injection (p.i.)) and $^{89}$Zr-labeled Nivolumab PET (37.09 MBq, 162 h p.i.) demonstrate heterogeneous tracer uptake within and between tumors. **e** Patient 3 with tumor PD-L1 expression < 1%: $^{18}$F-FDG PET (268 MBq) ($^{18}$F-FDG PET scan images of both patients were used from archival PET-scans) demonstrates high glucose metabolism in the left-sided tumor. $^{18}$F-BMS-986192 PET (214.62 MBq, 1 h p.i.) demonstrates low tumor tracer uptake. $^{89}$Zr-labeled Nivolumab PET (37.27 MBq, 160 h p.i.) demonstrates heterogeneous tracer uptake in the tumor

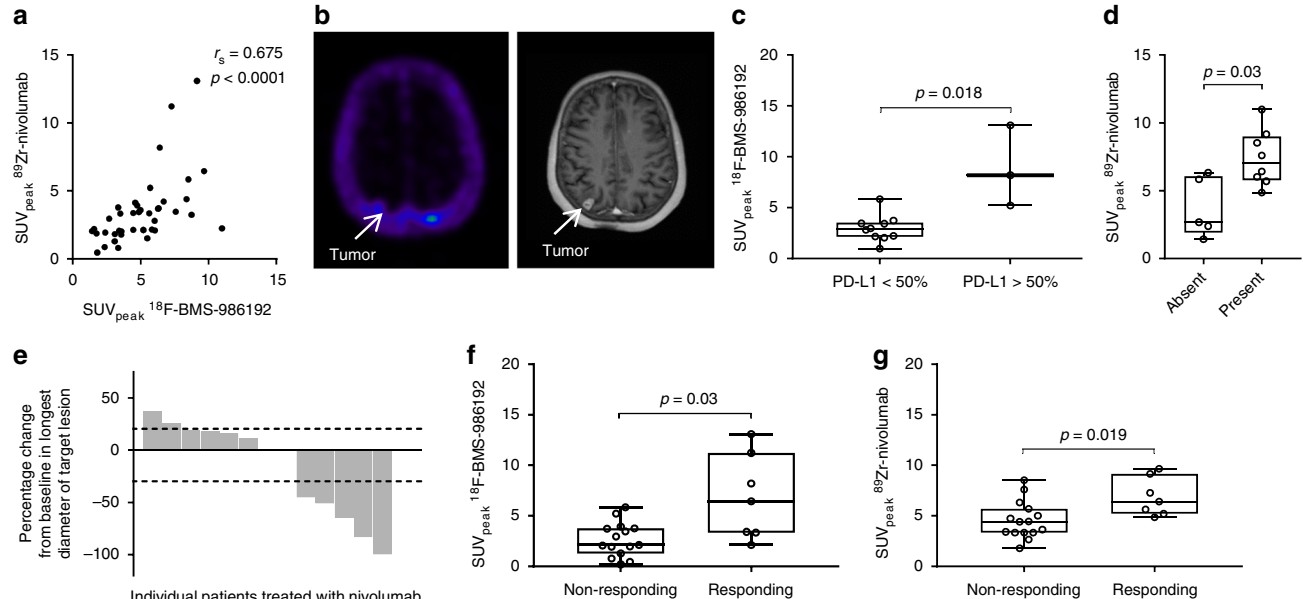

**Fig. 2** Tracer uptake and correlation with immunohistochemical staining and response. **a** Correlation between $SUV_{peak}$ for $^{89}Zr$-nivolumab and $^{18}F$-BMS-986192. $R_s = 0.68$, $p < 0.0001$ as determined by the Spearman rank correlation. **b** Low uptake of the $^{18}F$-BMS-986192 in an untreated brain metastasis in patient 4, due to PD-L1 expression heterogeneity or low CNS penetration of the tracer. **c** $^{18}F$-BMS-986192 $SUV_{peak}$ is higher in patients with ≥50% tumor PD-L1 expression. $p$-value is 0.018, as determined by the Mann–Whitney $U$-test. **d** Lesions with no PD-1 expression in aggregates have a lower $^{89}Zr$-nivolumab $SUV_{peak}$. $p$-value is 0.03, as determined by the Mann–Whitney $U$-test. **e** Waterfall plot of the best objective response according to RECIST 1.1. **f** $SUV_{peak}$ of the $^{18}F$-BMS-986192 tracer is higher in responding lesions as compared to non-responding lesions (comparison of lesions with diameter of 20 mm or more). $p$-value is 0.02, as determined by the Mann–Whitney $U$-test. **g** $SUV_{peak}$ of the $^{89}Zr$-nivolumab tracer is numerically higher in responding lesions (comparison of lesions with diameter of 20 mm or more). $p$-value is 0.019, as determined by the Mann–Whitney $U$-test. For all the boxplots, the lower edge of the box represents the first quartile and the upper edge represents the third quartile. The horizontal line inside the box indicates the median. Whiskers identify the minimum and the maximum value

**Heterogeneous radiotracer uptake.** Both tracers showed adequate tumor-to-background contrast for tumor visualization (Fig. 1d, e). Forty-five lesions were delineated with $^{18}F$-BMS-986192 and forty-two lesions with $^{89}Zr$-nivolumab (3.5 and 3.2 lesions per patient, respectively). SUV at 70–90 min p.i. was used to quantify $^{18}F$-BMS-986192 uptake as this semi-quantitative uptake parameter correlated optimally to the full kinetic model, derived from dynamic scanning (data not reported). Lesional uptake of $^{18}F$-BMS-986192 and $^{89}Zr$-nivolumab showed a positive correlation (Spearman rank correlation, $R_s = 0.68$, $p < 0.0001$), although this correlation was mainly driven by lesions with low uptake (Fig. 2a). $^{18}F$-BMS-986192 uptake heterogeneity was observed both between and within patients (Supplementary Fig. 1b). Two patients had untreated brain metastases (patients 4 and 10). Both tracers showed accumulation in selected, but not all, brain metastases in both patients. Lower SUV values ($SUV_{peak}$ range 0.2–0.9 for $^{18}F$-BMS-986192 and 0.8–2.4 for $^{89}Zr$-nivolumab) were observed in the brain metastasis as compared to the extracerebral lesions (Fig. 2b), which may be due to low CNS tracer penetration and/or variation in PD-1 and PD-L1 expression.

**Radiotracer uptake correlates with immunohistochemistry.** For non-CNS lesions, the median $^{18}F$-BMS-986192 $SUV_{peak}$ was higher for lesions with ≥50% tumor PD-L1 expression by IHC than for lesions with <50% expression (8.2 vs. 2.9, $p = 0.018$, Mann–Whitney $U$-test) (Fig. 2c). $^{89}Zr$-nivolumab uptake was higher in patients whose tumor biopsie(s) showed aggregates of PD-1[13] positive tumor-infiltrating immune cells (median $SUV_{peak}$ 7.0 vs. 2.7, $p = 0.03$, Mann–Whitney $U$-test) (Fig. 2d). The objective response rate in this study was 38% (5/13 subjects) (Fig. 2e). Of these five responders, one had a tumor biopsy with

PD-L1 IHC ≥1% but <50% (patient 4) and two with PD-L1 IHC ≥50% (patient 2 and 5). The other 2 responding patients were PD-L1 negative by IHC (patient 6 and 7). Tumor PD-L1 IHC did not correlate with response ($p = 0.7$ for ≥1% and $p = 0.06$ for ≥50%, Mann–Whitney $U$-test). PD-1 expression in aggregates by IHC was a predictor of response (p = 0.011, Mann–Whitney $U$-test).

**Radiotracer uptake is related to partial response.** Patients with a response after 3 months of nivolumab treatment showed a higher, although statistically not significant, $^{18}F$-BMS-986192 uptake in the biopsied tumor lesion than patients without a response (median $SUV_{peak}$ 4.3 vs. 2.2, $p = 0.089$, Mann–Whitney $U$-test) (Supplementary Fig. 2). Mixed responses were observed in 3 patients (patient 10, 11, and 12). To correlate the uptake of the radiotracers with response, we excluded the lesions with a diameter <20 mm, as the uptake in this lesions could be underestimated by partial volume effect. Response evaluation on a lesional basis (excluding lesions with a diameter less than 20 mm) showed that $^{18}F$-BMS-986192 $SUV_{peak}$ was higher for responding lesions than non-responding lesions (median 6.5 vs. 3.2, $p = 0.03$, Mann–Whitney $U$-test) (Fig. 2f), and an analogous lesional correlation was noted for $^{89}Zr$-nivolumab (median $SUV_{peak}$ 6.4 vs. 3.9, $p = 0.019$, Mann–Whitney $U$-test) (Fig. 2g).

## Discussion
In this study we showed that non-invasive quantification of PD-L1 and PD-1 is feasible with PET using the radiotracers $^{18}F$-BMS-986192 and $^{89}Zr$-nivolumab. Importantly, no severe tracer-related adverse events occurred during tracer injection and PET-CT scanning. Tumor uptake could be quantified in all patients, with substantial tracer uptake heterogeneity between

patients as well as within patients between different tumor lesions. Two patients with untreated brain metastases were included in this study, of which not all metastases were visible. This could be due to low CNS penetration of the tracers or to low PD-1 or PD-L1 expression.

$^{18}$F-BMS-986192 and $^{89}$Zr-nivolumab uptake correlated with PD-(L)1 expression by IHC in tumor tissue. This implies that these tracers could potentially be used for serial and non-invasively quantification of PD-(L)1 expression (dynamics) in clinical studies[14,15]. Of interest, a subset of tumors showed low PD-L1 expression by IHC in the biopsy specimen but relatively high SUV$_{peak}$ on the $^{18}$F-BMS-986192 PET-CT scan (e.g., patient 7). A possible explanation for this could be heterogeneity of PD-L1 expression in the lesion[16,17].

Although the relatively low spatial resolution of PET-CT, typically ~5 mm, limits detailed monitoring of intralesional uptake heterogeneity, the whole body character of PET-CT allows assessment of intralesional and interlesional uptake heterogeneity with the use of different uptake parameters, concentrating on the median value for a lesion (e.g., SUV$_{peak}$) or the higher or lower uptake voxels (e.g., SUV$_{max}$). PET imaging might identify a subgroup of patients that could respond to therapy, despite low PD-L1 expression in a small biopsy specimen (assuming lesions are large enough to be imaged by PET-CT and manifest high SUV). The correlation of lesion $^{18}$F-BMS-986192 SUV with PD-L1 expression and lesion $^{89}$Zr-nivolumab SUV with PD-1$^{+}$ lymphoid aggregates may also be useful to study drug induced changes in the tumor microenvironment in immunotherapy combination trials.

We found a relationship between SUV$_{peak}$ and response for both tracers, but it should be noted that the sample size in this study is small. Larger datasets are needed to validate our results. As $^{89}$Zr-nivolumab needs time to distribute in the human body because of its rather large molecular size, the optimal time point of PET-CT imaging post-injection is 5–7 days. With $^{18}$F-BMS-986192, same-day imaging is optimal (comparable to $^{18}$F-FDG PET-CT) and therefore more convenient to use in daily practice. However, larger datasets are needed to investigate the predictive value of $^{18}$F-BMS-986192 PET as a same-day, whole body, non-invasive biomarker of response to PD-(L)1 checkpoint inhibitor therapy.

In summary, this proof-of-principle study shows that in vivo molecular imaging of the PD-1/PD-L1 axis in NSCLC using $^{18}$F-BMS-986192, and $^{89}$Zr-nivolumab is feasible and safe in humans. $^{18}$F-BMS-986192 tumor uptake correlated with PD-L1 expression by IHC, and $^{89}$Zr-nivolumab uptake correlated with PD-1 expression on lymphocytic aggregates by IHC. This implies that these tracers could potentially be used to longitudinally and non-invasively quantify PD-(L)1 expression in future immunotherapy studies. Larger datasets are needed to validate these results.

## Methods

**Patient selection.** Between September 2016 and July 2017, patients with advanced NSCLC eligible for nivolumab treatment according to national and EMA guidelines were asked to participate in this study. Thirteen patients were included in this single-center, single arm, open-label, first-in-human, exploratory biomarker study. The study was approved by the local Institutional Review Board (Medical Ethics Committee of the VU University Medical Centre, Amsterdam). Written informed consent was obtained prior to study enrolment in all human participants and is conducted in accordance with the Declaration of Helsinki. The trial was registered at www.clinicaltrialsregister.eu (2015-004760-11).

Eligibility criteria were a histologically or cytologically confirmed diagnosis of stage IV, EGFR wild type and EML4-ALK fusion-negative NCSLC, measurable disease according to RECIST 1.1[18], ECOG performance status of 0–1, and a willingness to undergo a histological biopsy. Main exclusion criteria were symptomatic central nervous system (CNS) metastases and/or carcinomatous meningitis, use of corticosteroids with an equivalent of >10 mg prednisone, interstitial lung disease/active pneumonitis or active infection. Patients with asymptomatic central nervous system (CNS) metastases were allowed to participate.

**$^{18}$F-anti-PD-L1 PET-CT scan (PD-L1 PET).** $^{18}$F-BMS-986192 was synthesized at the GMP lab of the department of Radiology & Nuclear Medicine of the VUmc, Amsterdam, according to GMP guidelines. To a dried [$^{18}$F]fluoride a solution of BMT-180478 in DMSO is added. The reaction vessel is heated for 10 min at 120 °C. The reaction mixture is diluted with H$_2$O and purified over a Phenomenex Luna C18(2) 5 μm 250 × 10 HPLC column. The collected [$^{18}$F]BMT-187144 is trapped on a solid phase extraction (SPE) cartridge and eluted into the second reaction vessel with ethanol after which it is evaporated to dryness. At 45 °C [$^{18}$F]BMT-187144 is reacted with BMT-192920 for 45 min in PBS to form the final compound. The crude reaction mixture is purified by size exclusion using a PD-10 desalting column and PBS. The collected fraction containing the [$^{18}$F]BMS-986192 is sterile filtrated and dispensed using reduced pressure. Details can be found in supplementary data online (Supplementary Methods). Tracer injection was preceded by a low-dose CT (120 kV, 30 mAs) for attenuation correction and the CT served as an anatomical map. A dosimetry acquisition was done for the first patient. Following injection of 65.5 MBq, this patient underwent 4 consecutive PET-CT scans at $T = 1$ min, 35 min, 70 min and 105 min post-injection. In the subsequent 12 patients, the CT scan was followed by an injection of 3 MBq/kg $^{18}$F-BMS-986192 ±10% with a lower limit of 1.5 MBq/kg. The specific activity of the tracer was required to be equal or above 6.1 GBq/μmol. Immediately following injection, a 60-min dynamic PET-CT scan was conducted, followed by a 30-min static PET-CT scan (3 min per bed-position, 10–12 bed positions, depending on the length of the patient) covering the brain to the upper legs.

**$^{89}$Zr-nivolumab PET-CT scan.** $^{89}$Zr has been produced and purified[19] and is coupled to mAbs via the bifunctional chelate desferal (Df)[20,21], which has been safely used in the clinic before (for an overview see ref.[22]). $^{89}$Zr-Nivolumab is produced in compliance with current Good Manufacturing Practice at the VU University Medical Center. The procedures for radiolabeling nivolumab with $^{89}$Zr have been validated with respect to the final quality of the prepared conjugate. Details can be found in supplementary data online (Supplementary Methods). Nivolumab is labeled with $^{89}$Zr in an inert way, ensuring that $^{89}$Zr-Nivolumab pharmacokinetics equal that of un-labeled nivolumab kinetics. On day 2, patients were injected with $^{89}$Zr-nivolumab (37 MBq ± 10%, 2 mg nivolumab). For the first three subjects, static PET-CT scans were obtained 1 h and 3, 5 and 7 days p.i. for dosimetry purposes (data not reported), while the subsequent ten subjects were scanned 7 days p.i. due to optimal tumor-to-background contrast observed for this time point in the first three subjects. On day 12, patients received the first cycle of nivolumab treatment (3 mg/kg) and within two hours a second injection of $^{89}$Zr-nivolumab was administered. For the first three subjects static PET-CT scans were obtained 1 h and 3, 5 and 7 days p.i. for dosimetry purposes, all other patients received a single scan 7 days p.i. (data not reported). One patient received this scan at day 6 p.i. due to logistical reasons. The field of view (FoV) encompassed the brain to the mid-femur. A low-dose CT (120 kV, 30 mAs) was performed before every PET-CT scan for attenuation correction.

**PET-CT scan analyses.** The standardized uptake value (SUV) normalized to body mass in all organs that could be distinguished from background (liver, spleen, brain, lung, bladder, gall bladder, and kidneys) was determined using the mean activity concentration in volumes of interest (VOI) drawn over the entire organs using in-house developed software[23]. VOIs were drawn independently for each acquired PET image. Organs containing metastases were not used for analysis. Visual assessment of tumor uptake was performed by a nuclear physician and determined as focal uptake exceeding the local background. For the $^{18}$F-BMS-986192 PET-CT scans, VOIs were semi-automatically delineated. VOIs on the $^{89}$Zr-nivolumab PET-scans were determined and delineated manually on an attenuation corrected PET-CT scan. Activity concentrations (AC$_{peak}$) were derived per VOI.

**Adverse events.** Radiotracer related adverse events were recorded from the initial signing of the informed consent to the first full dose of nivolumab (i.e., 2 weeks after $^{18}$F-BMS-986192 injection).

**Treatment.** Patients were treated with 3 mg/kg nivolumab every two weeks until disease progression, unacceptable toxicity, or withdrawal of consent.

**Response evaluation.** Responding patients were defined as subjects with a confirmed partial (PR) or complete response (CR) after 12 weeks of treatment, according to RECIST 1.1. Responding lesions were defined as lesions with a reduction in size of ≥30% at 12 weeks after the start of nivolumab.

**Tumor biopsies.** Tumor biopsies were obtained before the start of the study. Nine of twelve patients received a fresh biopsy following prior therapy, while archival sections were used for three subjects. Tumor sample sections were stained with haematoxylin and eosin (H&E) and an experienced pulmonary pathologist (E.T.), blinded for clinical information, evaluated the slides. IHC was performed,

# ARTICLE

including staining for PD-L1 with the clinically validated DAKO 28.8 antibody[24] and PD-1 with Cell Marque Corporation Clone NAT105 antibody. PD-L1 expression was assessed as the percentage of tumor cells showing positive cell membrane staining in the sample. PD-L1 and PD-1 were also scored according to the SP142 system (IC0, IC1, IC2 to IC3)[13]. For this analysis, a binary system was used; PD-1 present (IC1 to IC3) or absent (IC0).

**Statistical analyses**. Statistical analysis were performed using SPSS Statistics for windows version 22.0. Comparisons between unpaired continuous groups were made using the Mann–Whitney $U$-test or $t$-test. The relationship between two variables was calculated using Spearman's rank correlation coefficient. Differences with a $p$-value of 0.05 or less were considered statistically significant.

## Data availability
The data that support the findings of this study are available within this article and its supplementary files or from the authors upon reasonable request.

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

## Author contributions
A.J.L. designed the clinical study and developed the protocol. E.F.S., G.A.M.S.D., A.D.W., E.T., and T.D.G. contributed to the concept of the study. A.N.N. and A.J.L. analyzed and interpreted the data and wrote the manuscript. D.Leung, R.S., S.D., W.H., P.M., D.Lipovsek, D.J.D, S.J.B., and L.M. contributed to the development of the [18]F-BMS-986192 tracer. D.Leung provided the figures of subject 2 and 3. M.C.H., R.B., and O.S.H. contributed to the data-analysis and interpretation. A.J.L., I.B., and E.F.S. recruited subjects for this study. A.P., D.J.V., N.H.H., and A.D.W. contributed to the fabrication of both tracers. E.T. performed and interpreted histopathological studies. A.N.N., A.J.L, M.C.H., I.B., O.S.H., E.F.S., E.T., and T.D.G. critically discussed the clinical results of the study. A.N.N., A.J.L, M.C.H., I.B, O.S.H., E.F.S., G.A.M.S.D., A.D.W., E.T., and T.D.G. contributed to the discussion of the data and reviewed all drafts of the manuscript.

## Additional information

**Competing interests:** We received a research grant for the implementation of this study from Bristol-Myers Squibb (BMS). D.Leung, R.S., S.D., W.H., P.M., D.Lipovsek, D.J.D, S.J.B., and L.M. are employees and stock holders of BMS. The remaining authors declare no competing interests.

