## [Peer Review File · Nature Communications]

Reviewers' comments:

Reviewer #1 :

(Remarks to the Author):

1. The manuscript does not clearly explain the time course of distribution for the F-18 labelled anti-PD-L1 agent. Is it eliminated completely from the body within 24 hours, or does it also redistribute over time (days) to tumor sites? The PET scanner cannot differentiate between the 18-F and Zr89 emissions, correct? If so, how do the investigators know that increasing SUV in tumors at day 7 is not a contribution from both the anti-PD-L1-F18 and Zr89-nivolumab?
2. Does dose of labelled compound affect subsequent imaging, for both kinetics and intensity of the SUV value? If the dose of labeled drug does not saturate binding sites in vivo including non-specific distribution, then amount of labelled compound reaching tumor may be delayed and diminished. The dose of the adnectin F-18 agent was not given. What studies were done to determine if the dose of labelled compound used in his trial was optimal for imaging (sensitivity and specificity)?
3. The following statement, 'Of these five responders, three had a tumor biopsy with PD-L1 IHC $\geq 1\%$ (patient 2, 4, 5) and two with PD-L1 IHC $\geq 50\%$ (patient 2 and 5). Please explain if patients 2 and 5 are designated as PD-L1 IHC $\geq 1\%$ (but less than 50%) or $\geq 50\%$.
4. From visual inspection of the supplementary figure 1b, there appears to be a correlation between lesion size and lesion SUV. Was this analyzed formally? The correlation with overall objective response for the F18-anti-PD-L1 might be stronger if limited to patients with at least one lesion that was at least 2cm in size.
5. Please identify the patient numbers for the three mixed responses.
6. It may be more useful in this paper to score PD-L1 IHC expression using both immune and tumor cells in the microenvironment.
7. The conclusions from this study, particularly correlation with objective response, could be strengthened by increasing the sample size.
8. Although it doesn't appear to be the case, does combining the PD-L1 and PD-1 imaging data for an individual lesion improve the predictive value for response?
9. The authors did not include any discussion of the data. For example, the limited data suggest that in a patient whose tumor is PD-L1 negative or low by IHC (for example patient #7), PET imaging could identify a subgroup that could respond to therapy (assuming lesions are large enough to be imaged by the PET technique and manifest high SUV). The correlation of lesion peak SUV with PD-L1 expression and PD-1+ lymphoid aggregates may be useful in the future (for example, one might select different approaches for PD-L1 hot tumors, versus PD-L1 'colder tumors' with high PD-1 SUV versus PD-L1 'cold tumors' with low PD-1 SUV).

Reviewer #2:

(Remarks to the Author):

The manuscript entitled "Whole body PD-L1 and PD-1 imaging using positron emission tomography in patients with non-small cell lung cancer treated with nivolumab" submitted by Niemeijer et al., to Nature communications (MS#NCOMMS-18-17979-T) is well written original clinical research communication. This is an important research, first-in human study showing tumor PD-L1 and PD-1 can be measured non-invasively using PET/CT in patients, to predict the anti-PD-L1 treatment responders and heterogeneity of the PD-L1 expression. Although development of these tracers was already published but using in multiple patients to measure both PD-L1 and PD-1 with the same patient is an interesting study, authors performed commendable research study. I think this pilot study will be very useful to the imaging investigators, to develop many more functional tracers.

Briefly, this clinical study detailing of measuring PD-L1 and PD-1 expression by in vivo PET in

whole body using two different tracers 18F-anti-PD-L1-Adnectin [BMS-986192] for PD-L1, and 89Zr-Nivolumab for PD-1, in multiple patients, in comparison with 18F-FDG PET. Authors were carefully designed the study and analyzed the data. Authors stated, that the results of in vivo imaging study data using 18F-BMS-986192 and 89Zr-nivolumab in tumor were correlated with IHC of both PD-L1 expression in tumor and PD-1 expression on lymphocytic aggregates, respectively. But there is no IHC data provided in this manuscript to validate this statement. Given this limited data, clinical imaging results of this study are encouraging to measure in vivo PDL-1/PD-1 expressions, but larger datasets are needed to further validate these results, to investigate the predictive value of 18F-BMS-986192 PET.

One drawback in the current form of the manuscript is that the organization of study method is a bit of confusing, for example which tracer first received by the patient PD-L1 or PD-1? instead of random order it should be presented more clear manner. I would suggest a simple scheme would be very helpful to follow the story. In addition, please provide IHC data for patient #2 corresponding with PET image for the comparison of PD-L1 expressions.

Overall, this proof-of-principle study shows that in-vivo molecular imaging of the PD-1/PD-L1 expression in NSCLC were feasible. I believe this data would very interesting to the chem. commun readers.

Comments

1. Please provide a clear study scheme (as a figure) to address the following
 - a. Patient imaging time points after each of the tracer?
 - b. In what sequence imaging was performed (PD-L1 first followed by PD-1)?
 - c. After therapy what is the time lag for imaging with respect to patient received treatment
 - d. All patients were received 18F-FDG PET scanning prior to new tracers imaging?
2. Figure 1, please include a sub caption on the PT/CT image, as PD-L1 image (left two images) and PD-1 image (right image), respectively.
3. Please include the information of administered dose and corresponding imaging time point e.g., 2h p.i. in the figure legend.
4. Patient#2, images were performed before treatment or post treatment or both?
5. Figure 2b, P value connector should be spread to include both data points (<50% and >50%).
6. Figure 2d, objective response data, accounted for only 11 patients, remaining two patients status?
7. Figure 2e, y-axis use the same tracer notation [18F-BMS-986192]
8. What was the amount of Nivolumab pre-dose administered, and how many hours prior to the 89Zr-Nivolumab tracer injection?
9. Line #53, There is no supplementary table 1c, I think it should be supplementary table 3
10. What was the specific activity of the 18F-BMS-986192 used in patients?
11. Please include the IHC data for patient #2 corresponding to Figure 1a.
12. Finally, authors should comment on expressions of PD-L1 and PD-1 correlation within the tumor, any correlation observed between these two check points would be valuable to the readers.

Reviewer #3:

(Remarks to the Author):

The study with radiotracers for labeling PD-1, such as zirconium-labeled nivolumab or the fluor-labeled anti-PD-L1 adnectin, is very innovative. The authors should state which of the 2 radiotracers would be the most convenient for further studies.

Some comments:

1. The pictorial representation of PD-L1 in the whole body is of great interest, specifically in

patients with low PD-L1 expression, and it seems that zirconium-labeled nivolumab represents the best method for identifying tumor heterogeneity. However, PD-L1 expression is under represented before treatment with immunotherapy. Following therapy with anti-PD-1 or anti-PD-L1 antibodies, TCR signaling is restored, releasing interferon and leading to upregulation of PD-L1 (Heim et al Cancer Res 2018). Therefore, the Zr-labeled nivolumab PET could be more demonstrative following one or two cycles of immunotherapy.

2. The authors should clarify the use of Zr-labeled nivolumab. If nivolumab blocks the expression of PD-L1 in the tumor tissue, how can this be reconciled with identification of PD-L1 by PET?

3. Some tumors also express glycosylated PD-L1 and, therefore, it may be less recognized. See: Li et al Cancer Cell 2018.

Reviewer #1 : (Remarks to the Author):

1. **Reviewer:** The manuscript does not clearly explain the time course of distribution for the F-18 labelled anti-PD-L1 agent. Is it eliminated completely from the body within 24 hours, or does it also redistribute over time (days) to tumor sites? The PET scanner cannot differentiate between the 18-F and Zr89 emissions, correct? If so, how do the investigators know that increasing SUV in tumors at day 7 is not a contribution from both the anti-PD-L1-F18 and Zr89-nivolumab?

Authors: We would like to thank the reviewer for his/her time to read the manuscript and for the valuable comments. We agree with the reviewer that a PET scanner cannot differentiate between fluor-18 and zirconium-89 emissions. However, fluor-18 has a half-life of approximately 109 minutes and will therefore not contribute to the increasing SUV in tumors at day 7.

2. **Reviewer:** Does dose of labelled compound affect subsequent imaging, for both kinetics and intensity of the SUV value? If the dose of labeled drug does not saturate binding sites in vivo including non-specific distribution, then amount of labelled compound reaching tumor may be delayed and diminished. The dose of the adnectin F-18 agent was not given. What studies were done to determine if the dose of labelled compound used in this trial was optimal for imaging (sensitivity and specificity)?

Authors: We thank the reviewer for raising this important issue. The dose in mg of both tracers (¹⁸F-BMS-986192 and ⁸⁹Zr-nivolumab) did not exceed 3 mg (in total, not per kg) per injection. Prior preclinical toxicity and dosimetry studies (Donnelly DJ, et al. J Nucl Med 2018) defined the optimal dose of ¹⁸F-BMS-986192 to be used in humans. In this study it was shown that co-administration of 1 mg/kg of unlabeled compound (BMS-986192) blocked PD-L1 and resulted in ~90% reduction of ¹⁸F-BMS-986192 radiotracer binding to near-background levels. Co-administration of a non-PD-L1 binding adnectin did not affect the binding of ¹⁸F-BMS-986192. Full kinetic modeling of ¹⁸F-BMS-986192 in humans was also done in this study and will be described in a separate manuscript.

3. **Reviewer:** The following statement, 'Of these five responders, three had a tumor biopsy with PD-L1 IHC $\geq 1\%$ (patient 2, 4, 5) and two with PD-L1 IHC $\geq 50\%$ (patient 2 and 5). Please explain if patients 2 and 5 are designated as PD-L1 IHC $\geq 1\%$ (but less than 50%) or $\geq 50\%$.

Authors: We are sorry for not being clear about this point, and adjusted this in the manuscript.

4. **Reviewer:** From visual inspection of the supplementary figure 1b, there appears to be a correlation between lesion size and lesion SUV. Was this analyzed formally? The correlation with overall objective response for the F18-anti-PD-L1 might be stronger if limited to patients with at least one lesion that was at least 2cm in size.

Authors: Although there seems to be a correlation between lesion size and SUV uptake, this is mainly in the smaller lesions and could be explained by partial volume effect. Excluding lesions less than 2cm in size, shows a stronger correlation between response in all lesions and SUV uptake ($p=0.03$ for ¹⁸F-BMS-986192 and $p=0.019$ for ⁸⁹Zr-nivolumab). We adjusted the outcome in the manuscript and inserted these figures in the manuscript.

5. **Reviewer:** Please identify the patient numbers for the three mixed responses.

Authors: We have inserted this in the manuscript. Patient 10, 11 and 12.

6. **Reviewer:** It may be more useful in this paper to score PD-L1 IHC expression using both immune and tumor cells in the microenvironment.

Authors: Because tumor PD-L1 expression is the most widely used tissue biomarker with the best predictive value for response during PD-(L)1 antibody treatment, the aim of this study was to correlate PD-L1 imaging with tumor PD-L1 expression.

However, we agree with the reviewer that the correlation between PD-L1 SUV and PD-L1 IHC expression using both immune and tumor cells is of interest. For this purpose PD-L1 on immune cells was scored according to the SP142 system (IC0, IC1, IC2 to IC3). However, this correlation was not significant, possibly due to the low number of subjects or the type of scoring system used.

7. **Reviewer:** The conclusions from this study, particularly correlation with objective response, could be strengthened by increasing the sample size.

Authors: We agree with the reviewer on this point. This study is a feasibility study with both tracers, aiming at proof-of-concept, safety, biodistribution, kinetic modelling and correlation with IHC. A future study increasing the sample size with the ¹⁸F-BMS-986192 tracer is planned aiming to include 80 subjects with NSCLC.

8. **Reviewer:** Although it doesn't appear to be the case, does combining the PD-L1 and PD-1 imaging data for an individual lesion improve the predictive value for response?

Authors: This is also an interesting comment, but unfortunately this was not the case, as might be expected in this small group.

9. **Reviewer:** The authors did not include any discussion of the data. For example, the limited data suggest that in a patient whose tumor is PD-L1 negative or low by IHC (for example patient #7), PET imaging could identify a subgroup that could respond to therapy (assuming lesions are large enough to be imaged by the PET technique and manifest high SUV). The correlation of lesion peak SUV with PD-L1 expression and PD-1+ lymphoid aggregates may be useful in the future (for example, one might select different approaches for PD-L1 hot tumors, versus PD-L1 'colder tumors' with high PD-1 SUV versus PD-L1 'cold tumors' with low PD-1 SUV).

Authors: Thank you for your comment, we share your thoughts and added text to the discussion section in the revised version of the manuscript.

Reviewer #2:

(Remarks to the Author):

The manuscript entitled "Whole body PD-L1 and PD-1 imaging using positron emission tomography in patients with non-small cell lung cancer treated with nivolumab" submitted by Niemeijer et al., to Nature communications (MS#NCOMMS-18-17979-T) is well written original clinical research communication. This is an important research, first-in human study showing tumor PD-L1 and PD-1 can be measured non-invasively using PET/CT in patients, to predict the anti-PD-L1 treatment responders and heterogeneity of the PD-L1 expression. Although development of these tracers was already published but using in multiple patients to measure both PD-L1 and PD-1 with the same patient is an interesting study, authors performed commendable research study. I think this pilot study will be very useful to the imaging investigators, to develop many more functional tracers.

Briefly, this clinical study detailing of measuring PD-L1 and PD-1 expression by in vivo PET in whole body using two different tracers ^{18}F -anti-PD-L1-Adnectin [BMS-986192] for PD-L1, and ^{89}Zr -Nivolumab for PD-1, in multiple patients, in comparison with ^{18}F -FDG PET. Authors were carefully designed the study and analyzed the data. Authors stated, that the results of in vivo imaging study data using ^{18}F -BMS-986192 and ^{89}Zr -nivolumab in tumor were correlated with IHC of both PD-L1 expression in tumor and PD-1 expression on lymphocytic aggregates, respectively. But there is no IHC data provided in this manuscript to validate this statement. Given this limited data, clinical imaging results of this study are encouraging to measure in vivo PDL-1/PD-1 expressions, but larger datasets are needed to further validate these results, to investigate the predictive value of ^{18}F -BMS-986192 PET.

One drawback in the current form of the manuscript is that the organization of study method is a bit of confusing, for example which tracer first received by the patient PD-L1 or PD-1? instead of random order it should be presented more clear manner. I would suggest a simple scheme would be very helpful to follow the story. In addition, please provide IHC data for patient #2 corresponding with PET image for the comparison of PD-L1 expressions.

Overall, this proof-of-principle study shows that in-vivo molecular imaging of the PD-1/PD-L1 expression in NSCLC were feasible. I believe this data would very interesting to the chem. commun readers.

Comments

1. **Reviewer:** Please provide a clear study scheme (as a figure) to address the following:
 - a. Patient imaging time points after each of the tracer?
 - b. In what sequence imaging was performed (PD-L1 first followed by PD-1)?
 - c. After therapy what is the time lag for imaging with respect to patient received treatment
 - d. All patients were received ^{18}F -FDG PET scanning prior to new tracers imaging?

Authors: We thank the reviewer for the careful evaluation of our manuscript and valuable comments. We inserted a figure in the revised manuscript addressing point a-c. Although most patients received an ^{18}F -FDG PET scan as part of their diagnostic work-up or response monitoring to prior treatment, an ^{18}F -FDG PET scan was not part of this study and not all patients received a ^{18}F -FDG PET scan prior to ^{18}F -BMS-986192 and ^{89}Zr -nivolumab scanning.

2. **Reviewer:** Figure 1, please include a sub caption on the PT/CT image, as PD-L1 image (left two images) and PD-1 image (right image), respectively.

Authors: We adjusted the figure.

3. **Reviewer:** Please include the information of administered dose and corresponding imaging time point e.g., 2h p.i. in the figure legend.

Authors: We included the information.

4. **Reviewer:** Patient#2, images were performed before treatment or post treatment or both?

Authors: Thank you for your comment, the images were performed before treatment.

5. **Reviewer:** Figure 2b, P value connector should be spread to include both data points (<50% and >50%).

Authors: We adjusted the figure.

6. **Reviewer:** Figure 2d, objective response data, accounted for only 11 patients, remaining two patients status?

Authors: Response evaluation could not be performed because of rapid deterioration and death short after 1 (patient 9) or 2 cycles of nivolumab (patient 1).

7. **Reviewer:** Figure 2e, y-axis use the same tracer notation [18F-BMS-986192].

Authors: Thank you for this comment. In the revised manuscript, we have changed the tracer notation.

8. **Reviewer:** What was the amount of Nivolumab pre-dose administered, and how many hours prior to the ⁸⁹Zr-Nivolumab tracer injection?

Authors: We performed two imaging acquisitions with the ⁸⁹Zr-nivolumab tracer, one with the tracer dose only (without co-administration with non-radiolabeled nivolumab) and one with co-administration of a therapeutical dose of 3 mg/kg non-radiolabeled nivolumab. This pre-dose was administered preferably directly prior to the tracer injection, with a maximum interval of two hours between the pre-dose and the radiotracer injection. In this manuscript we only described the scan results that were obtained without a pre-dose. In short, the results were not significantly influenced by the use of a pre-dose. This is described in the methods section.

9. **Reviewer:** Line #53, There is no supplementary table 1c, I think it should be supplementary table 3

Authors: Thank you for this comment. We adjusted this in the text.

10. **Reviewer:** What was the specific activity of the 18F-BMS-986192 used in patients?

Authors: All patients received a tracer dose with a specific activity of ≥ 6.1 GBq/ μ mol as this was required for the release of the tracer. The specific activity was higher than the one used in the previous study (Donnelly DJ, et al. J Nucl Med 2018). This is added to the methods section.

11. **Reviewer:** Please include the IHC data for patient #2 corresponding to Figure 1a.

Authors: We inserted the IHC data for patient 2 and 3 in the figure.

12. **Reviewer:** Finally, authors should comment on expressions of PD-L1 and PD-1 correlation within the tumor, any correlation observed between these two check points would be valuable to the readers.

Authors: Thank you for this valuable comment, we inserted the following sentence (Lesional uptake of ¹⁸F-BMS-986192 and ⁸⁹Zr-nivolumab showed a positive correlation ($R_s = 0.68$, $p < 0.0001$), although this correlation was mainly driven by lesions with low uptake.) to the research section and added a figure (2a) in the revised manuscript.

Reviewer #3: (Remarks to the Author):

The study with radiotracers for labeling PD-1, such as zirconium-labeled nivolumab or the fluorine-labeled anti-PD-L1 adnectin, is very innovative. The authors should state which of the 2 radiotracers would be the most convenient for further studies.

Some comments:

1. **Reviewer:** The pictorial representation of PD-L1 in the whole body is of great interest, specifically in patients with low PD-L1 expression, and it seems that zirconium-labeled nivolumab represents the best method for identifying tumor heterogeneity. However, PD-L1 expression is under represented before treatment with immunotherapy. Following therapy with anti-PD-1 or anti-PD-L1 antibodies, TCR signaling is restored, releasing interferon and leading to upregulation of PD-L1 (Heim et al Cancer Res 2018). Therefore, the Zr-labeled nivolumab PET could be more demonstrative following one or two cycles of immunotherapy.

Authors: We would like to thank the reviewer for the time to read the manuscript and the valuable comments. Heim et al. described very interesting details of T-cell activation after treatment with PD-(L)1 antibody therapy. We performed imaging of the PD-1 receptor status with ⁸⁹Zr-nivolumab and the PD-L1 receptor status with ¹⁸F-BMS-986192. Both tracers were injected and patients were imaged prior to nivolumab treatment. Thus the tracers were studied for their predictive value for outcome and correlation with tissue IHC prior to treatment. We agree with the reviewer that on-treatment imaging to evaluate the value of the tracers to image early IO induced changes on a tissue level is of utmost interest. However, this is beyond the scope of this manuscript.

2. **Reviewer:** The authors should clarify the use of Zr-labeled nivolumab. If nivolumab blocks the expression of PD-L1 in the tumor tissue, how can this be reconciled with identification of PD-L1 by PET?

Authors: Thank you for this comment. As nivolumab is an antibody against PD-L1, we believe that ⁸⁹Zr-nivolumab imaging represents the PD-1 receptor status on immune infiltrating cells. The ¹⁸F-BMS-986192 (anti-PD-L1) tracer was used to image the PD-L1 receptor status. ¹⁸F-BMS-986192 injection and imaging was performed more than 5 half-lives prior to the injection and imaging of ⁸⁹Zr-nivolumab. As both PD-L1 and PD-1 expression in the tumor microenvironment could be predictive for response to PD-(L)1 antibody treatment, we were interested in both the PD-1 as the PD-L1 receptor status.

3. **Reviewer:** Some tumors also express glycosylated PD-L1 and, therefore, it may be less recognized. See: Li et al Cancer Cell 2018.

Authors: We agree with the reviewer that Li et al. describe the interesting observation of glycosylation of PD-L1, thereby stabilizing PD-L1 and preventing its degradation. They also showed that monoclonal antibodies targeting glycosylated PD-L1 promote PD-L1 internalization and degradation. However, this research was preclinical and performed in triple negative breast cancer cells. Whether glycosylation of PD-L1 is abundant in NSCLC and what its role in NSCLC is with respect to immune suppression and response to PD-(L)1 antibody treatment is unknown to the best of our knowledge. We agree with the reviewer that it is expected that ¹⁸F-BMS-986192 does not image glycosylated PD-L1.

Best regards,

Dr. A.J. de Langen

REVIEWERS' COMMENTS:

Reviewer #1 (Remarks to the Author):

None

Reviewer #2 (Remarks to the Author):

The revised manuscript entitled "Whole body PD-L1 and PD-1 imaging using positron emission tomography in patients with non-small cell lung cancer treated with nivolumab" submitted by Niemeijer et al (NCOMMS-18-17979A) is looks good. In this revised manuscript authors addressed all questions/comments raised by the reviewers. Please accept the revised manuscript. Thanks.

Reviewer #3 (Remarks to the Author):

The authors have satisfactorily answered all queries and the study looks flawless and novel for the readers.

Rebuttal 2: REVIEWERS' COMMENTS: Reviewer #1 (Remarks to the Author):

None

Reviewer #2 (Remarks to the Author): The revised manuscript entitled “Whole body PD-L1 and PD-1 imaging using positron emission tomography in patients with non-small cell lung cancer treated with nivolumab” submitted by Niemeijer et al (NCOMMS-18-17979A) is looks good. In this revised manuscript authors addressed all questions/comments raised by the reviewers. Please accept the revised manuscript. Thanks.

Reviewer #3 (Remarks to the Author): The authors have satisfactorily answered all queries and the study looks flawless and novel for the readers.

Authors: We would like to thank all reviewers for their kind words after revision.